# Evolutionary history of host trees amplifies the dilution effect of biodiversity on forest pests

**Andrew V. Gougherty**[1,2]*, **T. Jonathan Davies**[2,3]

**1** USDA Forest Service, Northern Research Station, Delaware, Ohio, United States of America,
**2** Department of Botany, University of British Columbia, Vancouver, Canada, **3** Department of Forest &
Conservation Sciences, University of British Columbia, Vancouver, Canada

* andrew.gougherty@usda.gov

Evolutionary history of host trees amplifies the
dilution effect of biodiversity on forest pests. PLoS
Biol 22(2): e3002473. https://doi.org/10.1371/
journal.pbio.3002473

University, UNITED STATES

**Data Availability Statement:** Data used in this
study are available at the cited sources, and at doi.
org/10.6084/m9.figshare.21317979.

**Funding:** This project was funded by a Discovery
Grant from the Natural Sciences and Engineering

## Abstract

Biodiversity appears to strongly suppress pathogens and pests in many plant and animal systems. However, this "dilution effect" is not consistently detected, and when present can vary strikingly in magnitude. Here, we use forest inventory data from over 25,000 plots (>1.1 million sampled trees) to quantify the strength of the dilution effect on dozens of forest pests and clarify why some pests are particularly sensitive to biodiversity. Using Bayesian hierarchical models, we show that pest prevalence is frequently lower in highly diverse forests, but there is considerable variability in the magnitude of this dilution effect among pests. The strength of dilution was not closely associated with host specialization or pest nativity. Instead, pest prevalence was lower in forests where co-occurring tree species were more distantly related to a pest's preferred hosts. Our analyses indicate that host evolutionary history and forest composition are key to understanding how species diversity may dilute the impacts of tree pests, with important implications for predicting how future biodiversity change may affect the spread and distribution of damaging forest pests.

## Introduction

A better understanding of the role of biodiversity in moderating pathogen and pest prevalence is urgently needed as human activities continue to simplify the biotic landscape and reduce diversity at global scales [1] while also promoting the spread of many pest species beyond their native ranges [2]. The link between plant pests and biodiversity is of both ecological and economic importance as it may affect the management of agricultural/forestry species, shape biodiversity conservation policy (i.e., shifts towards low-intensity agroecosystems and enhanced agrobiodiversity), and inform the debate on land sharing versus land sparing [3]. While most recent meta-analyses have, generally, found support for the dilution effect, numerous case studies and systematic reviews have shown the effect is not universal, and that the strength of the dilution effect can vary among studies and with the ecology of the pathosystem. In forests in particular, numerous studies have shown support for the dilution effect on pest damage and occurrences [4–6], while others have found pests to be positively correlated with

Research Council of Canada to TJD. The funders had no role in study design, data collection and analysis, decision to publish, or preparation of the manuscript.

**Competing interests:** The authors have declared that no competing interests exist.

[7–9], or independent from [10,11] tree diversity. Thus, while studies have frequently found evidence in support of the dilution effect, we still lack robust generalizable models that allow us quantify the strength of the dilution effect across less well-studied pest systems that also account for potentially confounding factors, including climate, pest host ranges, and forest composition.

Responses of tree pests to biodiversity are likely influenced by both intrinsic pest traits and extrinsic traits of the forests where tree hosts occur. Pest host range, nativity, and taxonomic membership, for example, could each impact the likelihood of pests encountering and successfully utilizing hosts [12]. Monophagous pests may be the most impacted by biodiversity, as greater diversity will necessarily reduce the relative frequency of suitable hosts. However, generalists could also be negatively impacted if they are local specialists or have strong host preferences. The likelihood of pests encountering competent hosts will also be affected by pest dispersal mode. For example, passively dispersed fungal pathogens, which can produce many aerially dispersed spores, may be more likely to encounter hosts in diverse forests compared to actively dispersing insect pests, some of which rely on chemical signals and/or visual cues to locate hosts [12]. Pest nativity could additionally influence pest prevalence if native trees lack coevolved defenses to non-native pests, and absence of natural enemies might enable non-native pests to obtain higher population abundances [13].

Forest composition is another important factor regulating pest responses to biodiversity. Tree communities composed of species from diverse lineages, for example, may more often inhibit pest attacks than communities composed of an equivalent number of closely related species [14]. While there is some evidence that forest phylogenetic diversity may impact pest responses to biodiversity [12], this has typically been inferred from comparisons of "mixed" and "monospecific" forest stands. Such comparisons may elucidate the aggregate effects of tree phylogenetic diversity, but are unable to identify which pests may be diluted by which forests.

Here, we used Bayesian hierarchical modeling to explore evidence for the dilution effect of biodiversity on forest pests across the United States, utilizing a dataset of >25,000 systematically sampled forest plots (Fig 1). Previous analyses of forest inventory data to test for the dilution effect (e.g., [15]) have, critically, lacked information on pest occurrences within plots, and thus provide only limited inference on pest prevalence. Here, we extracted high-resolution, tree-level pest association data to quantify pest prevalence, and our analyses represents one of the largest empirical tests of the dilution effect in forest trees. We show that the dilution effect is common among tree pests, even after accounting for host abundance, climate, and pest nativity. However, the magnitude of dilution is mediated by the relative phylogenetic composition of forests, such that a given tree community may have a strong dilution effect on some pests, but not others, contingent on the evolutionary relationships between the forest tree community and a given pest's preferred hosts. Our results help explain why empirical evidence supporting the diluting effect of biodiversity on plant pests and pathogens has appeared mixed, and allow us to better identify regions most at risk from future pest outbreaks.

## Materials and methods

### Forest inventory data

We used publicly available forest inventory data to assess the role of tree diversity on pest prevalence. As part of continual surveys of forested lands of the United States, US Forest Service personnel record whether trees have visual symptoms of damage from a wide range of pest species. Up to 3 damaging agents (which can also include damage caused by mammals, as well as abiotic damage such as wind or ice) are recorded per tree, dependent on damage-specific reporting thresholds. Damaging agents are identified in the 3 DAMAGE_AGENT_CD

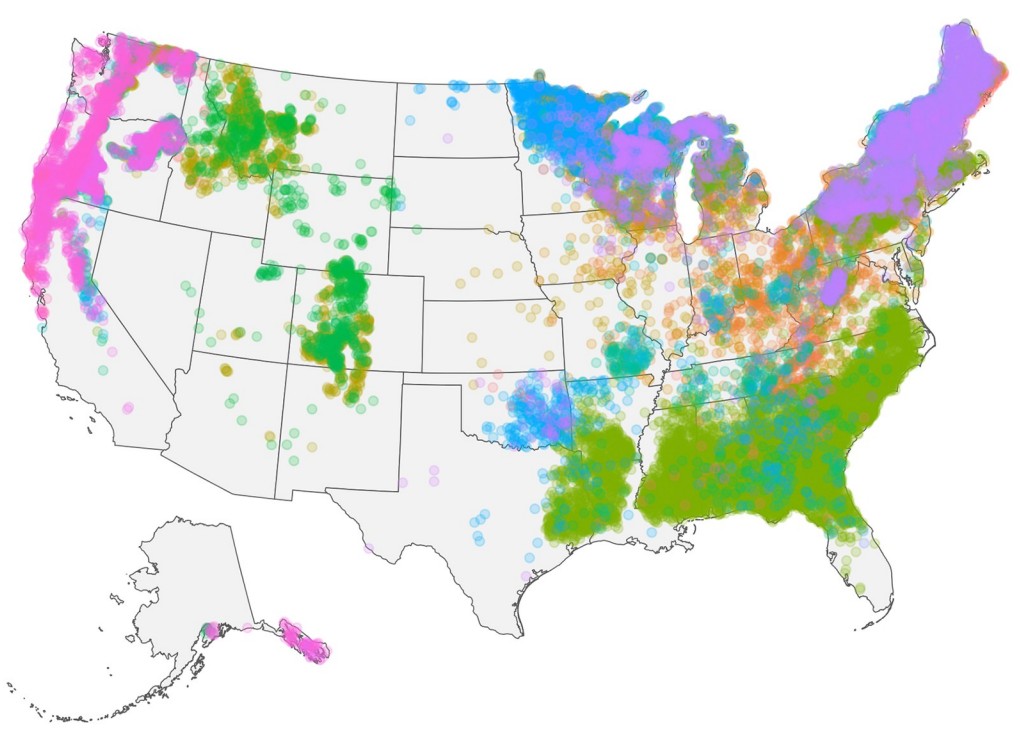

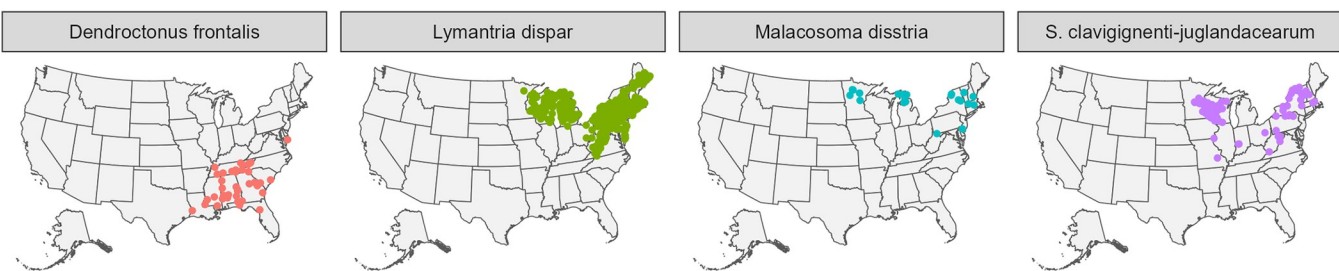

**Fig 1. Geographic distribution of the 26,555 forest inventory plots used here to assess the relationship between pest occurrences and diversity.** Unique colors represent 60 different pest species. Lower panels show 2 pests least affected by forest diversity, southern pine beetle (*Dendroctonus frontalis*), spongy moth (*Lymantria dispar*), and 2 pests most affected by forest diversity, forest tent caterpillar (*Malacosoma disstria*), and butternut canker (*Sirococcus clavigignenti-juglandacearum*). See also Figs 3 and 4. Base map is from US Census Bureau [https://www.census.gov/geographies/mapping-files/time-series/geo/cartographic-boundary.html]. Data underlying this figure can be found at doi.org/10.6084/m9.figshare.21317979.

columns in the tree-level database available from the Forest Inventory and Analysis (see the FIA database user guide [https://www.fia.fs.usda.gov/library/database-documentation/] for a full description of how plots and damaging agents are sampled). The Pacific Northwest Research Station records an additional (up to) 3 damaging agents per tree (located in the DMG_AGENT_CD_PNWRS column). These were also included in our analyses, removing duplicated data in any of the damaging agent columns. Although some trees could, hypothetically, have been subject to more than 3 damaging agents, Randolph and colleagues [16] previously reported that very few individual trees (typically 1% to 2% across species) had 3 reported damaging agents, and an even smaller fraction (<1%) would be subject to 4 or more damaging agents. These data thus include the relevant damage for the vast majority of trees within the FIA sampling scheme.

While the damage data from the FIA provides good geographic and taxonomic coverage of pests in the US, there are important limitations to these data [16]. First, surveyors are limited to visual, nondestructive sampling, and thus likely to only record instances that are readily perceived as causing tree injury. Second, because pests can cause similar types of damage, it is possible causative agents may be misidentified. However, surveyors receive extensive training and have the option, when damage is observed, but not identifiable to species, to record the damaging agent as a broader functional group (e.g., "boring insect" or "defoliator" as opposed to a specific species). Additionally, surveyors only record species-specific damage for those species that are known to be locally relevant, which helps ensure taxonomic consistency across the US. For example, southern pine beetle (*D. frontalis*) damage is only recorded in states covered by the Southern Research Station, while laminated root rot (*Phellinus weirii*) is only recorded in states covered by the Pacific Northwest Research Station. Third, FIA plots only encompass a small portion of the forested area in the US, and surveys are conducted at a particular point in time during the year such that small scale, brief, or otherwise acute outbreaks may be missed in any given year. Despite these caveats, these survey data represent some of the highest quality and most rigorously inventoried data on forest trees pests and have been used in numerous regional summaries (e.g., [17,18], see also [16]).

In each plot with available pest data, we calculated the number of trees recorded as being affected by each insect, pathogen, and parasitic plant damaging agent. Because surveys are limited to visual symptoms, we aggregated pests to the species-level, but this only affected one pest (*Cronartium quercuum f. sp. fusiforme*). For consistency, we verified that the tree with the recorded damage was a known host (described below) of the pest, and if not, it was excluded from the final dataset. In total, this step removed approximately 1% of the attacked trees in the dataset from the pest prevalence calculation (1,076 trees of the 107,108 originally included). While we acknowledge it is possible that pests may spillover to non-host trees, more often such observations likely represent mistaken assignments of the causative agent. In total, we assessed the prevalence of 60 pests on 1,160,522 unique trees across 26,555 plots, for a total of 30,727 pest × plot locations.

## Pest–host associations

For each pest species, we assembled a list of their global hosts by searching regional and global databases, including the USDA Fungal-Host database [19], CABI Crop Protection Compendium [20], North American Bark Beetle checklist [21], EPPO Global database [22], Leafroller Food Plant Database [23], World Lepidoptera Hostplants database [24], ScaleNet [25], the primary literature [26,27], and the gray literature [28,29]. Host names were standardized using the GBIF backbone with the rgbif package [30] in R. We retained only hosts that were identified to at least the species-level in the GBIF backbone.

## Statistical models

We used 2 alternative approaches for assessing the dilution effect of tree diversity on pest prevalence, first we fit a Bayesian hierarchical regression model, and second we examine causal pathways using a structural equation model (SEM). For both approaches, we modeled pest prevalence as the number of attacked hosts per pest within plots, divided by the total number of trees within the plot. This metric aligns with the definition of pathogen "incidence" in the plant pathology literature and "prevalence" in entomology and animal pathology literature [31]. For simplicity, we refer to this metric as "prevalence," although we appreciate that there are other definitions of pathogen prevalence in the literature. Both the Bayesian model and SEM included metrics of diversity (Shannon diversity and the proportion of trees within plots

that are known hosts) and climate variables strongly linked to productivity (mean temperature and precipitation). Pest-level metrics included the number of hosts associated with a given pest, as a metric of pest generalism/specialism, and the mean evolutionary distance between known hosts and all trees found within plots. The greater flexibility of the Bayesian model additionally allowed us to explore the effects of diversity on both the mean and per-pest prevalence, and thus also included pest type (e.g., pathogen, insect, parasitic plant) and pest nativity (native, non-native).

Shannon diversity was calculated using the "diversity" function in the vegan package [32] in R. We also examined Simpson's index, inverse Simpson's, and species richness, but found them all strongly correlated with Shannon diversity ($r = 0.96, 0.92, 0.88$), indicating that these metrics contain much of the same information. Proportion of hosts was calculated as the abundance of hosts within a plot divided by the total tree abundance within the plot. Evolutionary distance between known hosts and the trees within plots was calculated as the abundance-weighted average cophenetic distance between tree sets. This variable quantifies whether within-plot tree communities tend to be closely or distantly related to the known hosts of pests. The phylogenetic composition of host communities has been shown to be an important constraint on pest prevalence [14]. We here consider only the tree community composition as our focus is on pests that primarily affect tree species; however, for more generalist pests that can infest a variety of plant types (e.g., trees, understory shrubs, and herbaceous plants), it would be possible to expand this metric of evolutionary diversity to capture other elements of the plant community. Evolutionary distances were drawn from a phylogenetic tree, composed of all host species plus the tree species recorded within any plot, generated using the phylo. maker function in the V.PhyloMaker package [33], which prunes a megatree, based on Zanne and colleagues [34], to our species of interest and binds unsampled species to the inclusive genus or family.

Annual mean temperature (bio1) and precipitation (bio12) were downloaded from the WorldClim dataset at a resolution of 10 arc-minutes and extracted at plot locations [35]. Predictor variables demonstrated only moderate covariation across plots (strongest $r = 0.59$, between annual temperature and precipitation).

The Bayesian model was fit with Stan [36] called from brms [37] in R, assuming a beta distribution, and 4 Monte Carlo Markov chains each with 5,000 iterations, with the first 1,000 used as warmup. To fulfill the requirements of a beta distribution, any prevalence values of 1.0 were set to 0.99. All continuous variables were scaled to a mean of 0 and standard deviation of 1.0 to facilitate parameter comparisons. Pest identity was included as a grouping variable in the model which allowed us to estimate pest-level effects on Shannon diversity and relative host abundance. Model convergence was verified by checking rhat was near 1.0 and bulk and tail effective sample sizes (Bulk-ESS and Tail-ESS, respectively) were sufficiently large.

The large size of the dataset precluded us from including a spatial effect term directly into the model. We therefore examined spatial autocorrelation in both pest prevalence and in model residuals for 1,000 randomly selected plots, iterated 10 times (similar to [38]). Evidence for spatial autocorrelation was weak and not a statistical concern for either measure (S1 Fig).

The SEM was fit using least-squares models with the piecewiseSEM function in the psem package [39] in R. The variables included were identical to those in the Bayesian model, but also included effects of temperature and precipitation on forest diversity, and an effect of diversity on the proportion of hosts within plots. Pest-specific categorical variables were not included. The advantage of the SEM is that it allows us to more explicitly examine the indirect effects on aggregate pest prevalence. However, the Bayesian model allows us to estimate pest-specific slopes, and thus provides greater interpretability of pest-level effects, which we believe are critical for understanding the variability in the diluting effects of biodiversity. We therefore

focus on results from these models in the Results and Discussion unless specifically noted otherwise.

## Results

### Pest prevalence and tree diversity

Most pests were rare within plots. Across all 30,727 pest × plots combinations, there were on average (median) 42 trees (±19.7 SD, range: 1–175) within plots, of which 2 (±4.92 SD, range: 1–81) were reported to have pest damage—although some pests reached very high abundance, notably those affecting Pinaceae. Mountain pine beetles (*Dendroctonus ponderosae*) and western spruce budworm (*Choristoneura freemani*), for example, occasionally impacted >50 tree hosts within plots. Some pests also reached high relative abundance, attacking all trees within plots, including pine-oak rust (*Cronartium quercuum*), western spruce budworm, pine-pine gall rust (*Peridermium harknessii*), and emerald ash borer (*Agrilus planipennis*). There was similarly large variability in the number of plots where pests were reported. Frequency of infestation for a given pest was reported on average (median) in 95.5 (±1,182.74 SD) of the total 26,555 plots. *C. quercuum* was reported in, by far, the greatest number (8,205) of plots, followed by *Nectria coccinea* (3,217 plots), and *Glycobius speciosus* (2,149). Eight pests were recorded in only 1 plot, and a substantial number of pests (19) were reported in fewer than 10 plots. By partially pooling across pests, our Bayesian regression mixed model allows robust inference with unequal group sizes and additionally allows us to propagate uncertainty in parameter estimates, as captured in the credible intervals [40].

We found strong evidence for a dilution effect, whereby pest prevalence tends to be lower in plots with high tree diversity and low host abundance. These dilution effects were evident in both the SEM (S2 Fig) and in the global mean effect of the Bayesian model (Fig 2), as well as for nearly all pests individually (Figs 3 and S3 and S4). Surprisingly, neither host specialization nor nativity had a strong effect on pest prevalence, although the mean effect indicated that specialists had a tendency towards higher overall prevalence than generalists. Inspection of individual pest slopes revealed that, while several of the pests strongly diluted by diversity were specialists (e.g., *Sirococcus clavigignenti-juglandacearum*, *Hydria prunivorata*), and several pests least impacted by diversity were generalists (e.g., *L. dispar*, *Phytophthora cinnamomi*), there were multiple exceptions. For example, southern pine beetle (*D. frontalis*), a conifer specialist, showed little evidence of dilution, and even trended towards amplification. Likewise, the generalist forest tent caterpillar (*M. disstria*), a native insect that attacks a wide range of angiosperms and gymnosperms, tended to be diluted by diversity, possibly reflecting regional host preferences (Fig 4) [41]. Native and non-native pests were similarly distributed across the dilution-amplification gradient; however, it is notable that 5 of the 6 pests with the strongest tendencies toward dilution were native to forest stands (Figs 3 and 4).

### Importance of tree relatedness and host preferences

While the relationship between pest prevalence, tree diversity, and relative host abundance supported a general dilution effect, the evolutionary distance between pest hosts and the composition of tree species within plots had an important modifying effect (S5 Fig). Pest prevalence was higher in plots with tree communities more closely related to the known host trees of a given pest—suggesting that spillover to distantly related trees tends to dilute pest pressure. This phylogenetic effect implies that not all tree species contribute equally to dilution, and it is their evolutionary relatedness to known hosts that is important. The negative effect was also apparent in the SEM, despite it not including information on pest identity. The strength of this effect was equal to or greater than effects of climate, which indicated that pests tended to have

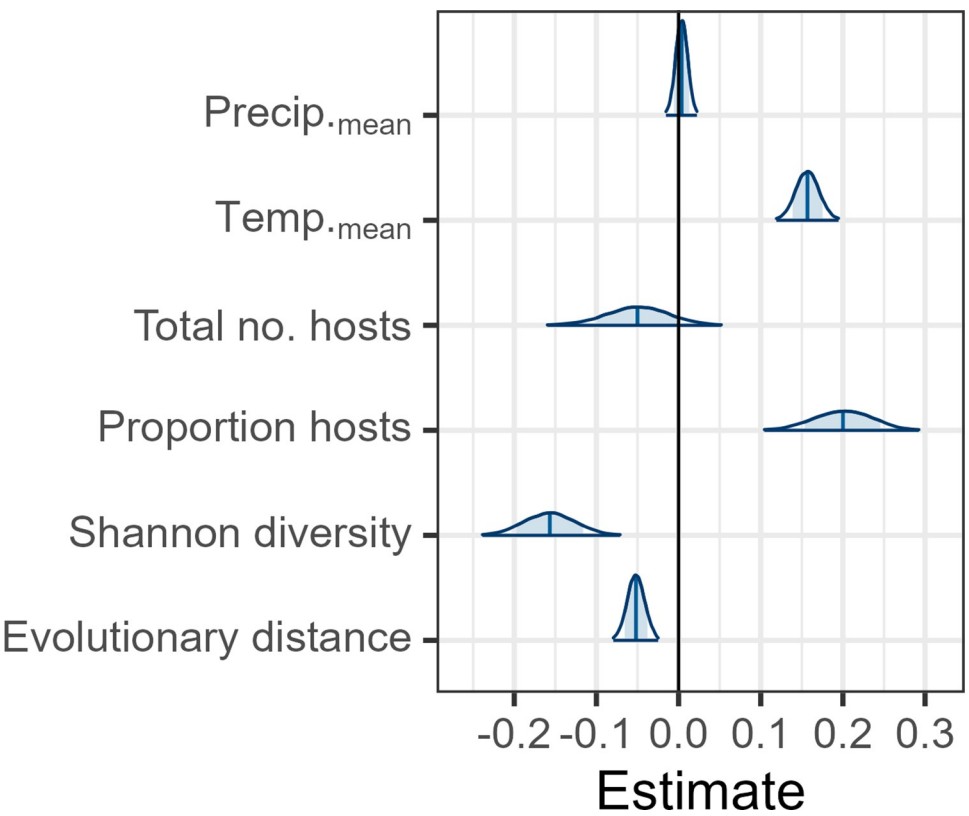

**Fig 2. Posterior parameter distribution from Bayesian regression model of pest occurrences.** Variables were scaled (mean of 0 and SD of 1.0) before model fitting. Blue vertical lines are means, shaded areas are 80% of draws, and distributions show 99% of draws. Categorical variables (pest type and nativity) are reported in S1 Table. Data underlying this figure can be found at doi.org/10.6084/m9.figshare.21317979.

greater prevalence in warmer climates, consistent with global biodiversity gradients. Precipitation gradients had a minimal impact on pest prevalence (Fig 2).

## Discussion

### Roles of phylogeny, host range, and pest nativity

The diluting effect of biodiversity on the prevalence of forest pests that we detect supports a growing body of literature on the relationship between host diversity and disease risk (but see Liu and colleagues [10], who found no significant negative effect of species richness on disease in forest ecosystems). However, our analysis is among the first to capture the variability in how individual pests differ in their response along biodiversity gradients and to demonstrate the modifying effect of the phylogenetic composition of host communities. While, in some systems, host abundance can facilitate the dilution effect independent from taxonomic richness or diversity per se [12,42], our results indicate both host diversity and abundance play a role in limiting pest prevalence. High forest tree diversity and low host abundance likely act in concert to reduce pest prevalence by decreasing encounter rates between pests and hosts, providing more suitable habitat for pest natural enemies, and presenting smaller "targets" for dispersing pests, decreasing the likelihood of pest establishment. These mechanisms, however, are magnified when forest trees are distantly related to pest hosts—consistent with the possibility that a single forest could simultaneously dilute one pest and amplify another.

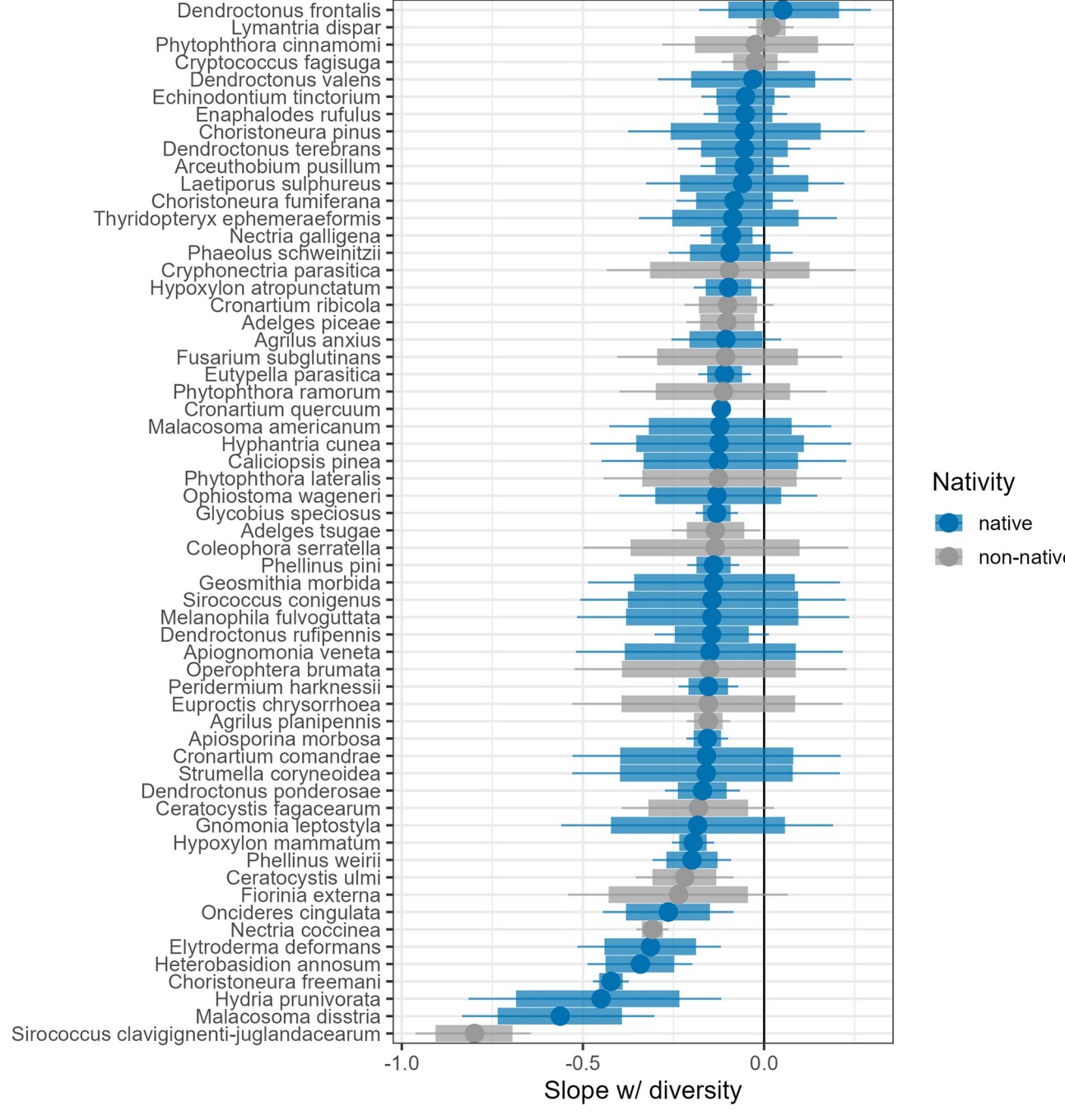

**Fig 3. Slope estimates between plot diversity and pest occurrences for all individual pests from our Bayesian regression model.** Negative values indicate a tendency towards dilution and positive values indicate a tendency towards amplification. Points and credible intervals are colored by pest nativity. Points are medians, shaded segments are 80% credible intervals, and lines are 95% intervals. See also S4 Fig. Data underlying this figure can be found at doi.org/10.6084/m9.figshare.21317979.

Our work provides little evidence for an effect of nativity on pest responses to biodiversity, even after adjusting for host breadth and forest community composition. The intrinsic and extrinsic factors affecting pest responses to biodiversity thus appear similar for native and non-native species. We did, however, find that non-native pests had higher prevalence overall

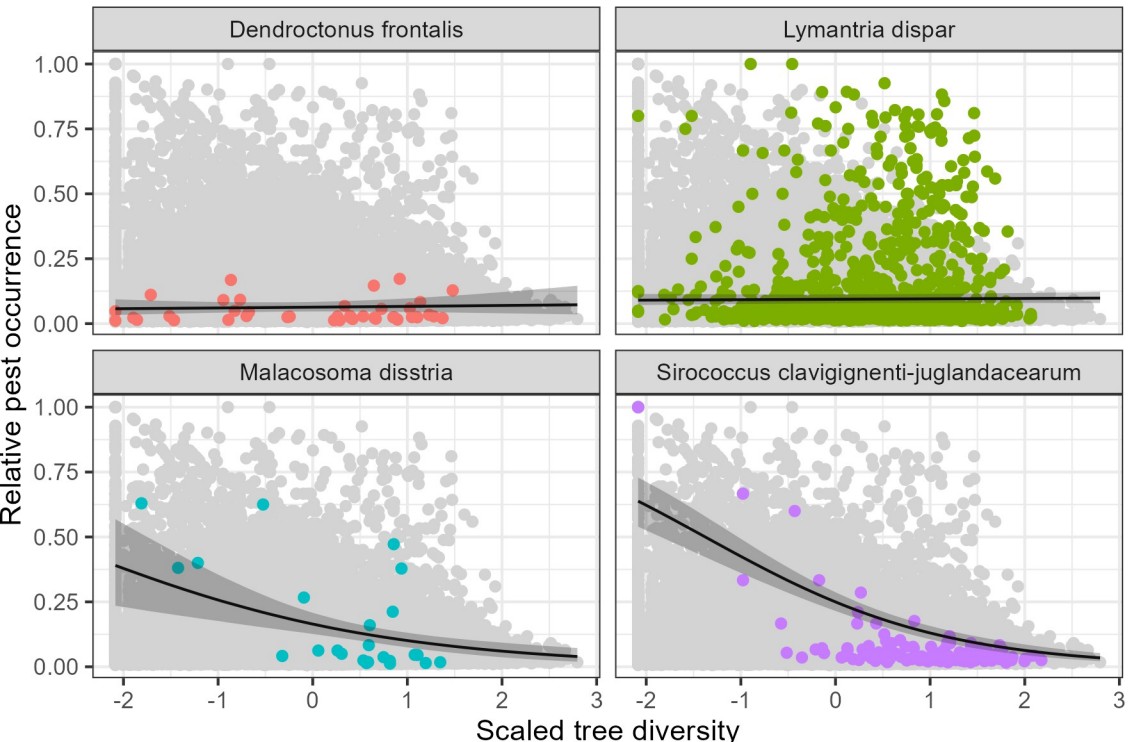

**Fig 4. Conditional effect, from our Bayesian regression model, of tree diversity on the relative abundance of 4 forest pests highlighting the variation in pest responses to biodiversity.** Pests include southern pine beetle (*D. frontalis*; native), spongy moth (*L. dispar*; non-native), forest tent caterpillar moth (*M. disstria*; native), and butternut canker (*Sirococcus clavigignenti-juglandacearum*; non-native). Gray points show data from all pests used in our model. Data underlying this figure can be found at doi.org/10.6084/m9.figshare.21317979.

compared to native pests. This is perhaps not surprising as hosts of non-native pests often have little coevolved resistance, and non-native pests may have fewer top-down controls (e.g., natural enemies), both of which could allow non-native pests to achieve higher prevalence. The severity of pest impacts on host trees could also affect their relative prevalence. Maladaptive hyper-virulence, for example, whereby pests quickly kill hosts, could theoretically increase pest prevalence in the short term but reduce pest prevalence in the long term, as available living hosts become rarer. While non-native pests may be most likely to be out of adaptive equilibrium with their hosts, there is as yet only limited evidence that they have systematically higher impacts on hosts than native pests [43,44].

Somewhat surprisingly, our expectation that specialist pests would more often tend to be diluted and generalist pests more often amplified by diversity was also not supported. There are various possible explanations for why we did not observe a strong effect of host breadth on dilution. For example, if the dominant tree species in a region are few and closely related, and the pest is a specialist on one or more of these dominant trees, then the addition of a host from the regional species pool is less likely to have a strong diluting effect. In our dataset, the southern pine beetle (SPB, *D. frontalis*), which specializes mostly on *Pinus*, provides one such example. Forests in the southern US, where SPB occurs, are dominated by numerous pine species (e.g., *Pinus palustris*, *P. taeda*, *P. echinata*, *P. elliottii*), all of which are hosts to SPB. A diverse tree community in the southern US, hence, is likely to contain numerous pine species which are all competent hosts for the SPB. Conversely, generalist pests may be diluted by diversity if hosts vary widely in their competence. *Phytophthora ramorum*, a generalist pathogen and

causal agent of Sudden Oak Death, for example, has been shown to be diluted by diversity [45]. Importantly, however, while a dilution effect may be observed at the community level, the probability of particular host species being infected by *P. ramorum* may be independent of diversity or even show the opposite relationship—being amplified by diversity rather than diluted [46]. This may be the case when a single, highly competent host tree dominates pest dynamics in the community.

In addition to information on host competence, other community-level data could help improve our understanding of pest prevalence. Functional traits have been linked to total pest loads and, when aggregated to the community, could affect the likelihood of pests establishing in a forest. For example, the proportion of fast-growing weedy species, which trade defense for growth, may be more likely to amplify pests compared to forests made up of slow-growing species that invest in various defense strategies—as individual species growth rates may be positively associated with herbivory [47]. Total forest productivity, leaf area, or functional type (angiosperm versus gymnosperm), among other forest characteristics, could each affect forest susceptibility to pest attack. Many such traits are phylogenetically conserved [48], and so could be partially captured by our evolutionary distance metric, but future work identifying landscape-level effects could be particularly informative for understanding pest prevalence and how they may shift with future landscape change.

The importance of the evolutionary distance between hosts and forest composition in mediating the dilution effect supports the notion that not all tree species contribute equally to dilution and helps explain the variability in the strength of dilution observed across pest species. This phylogenetic effect likely reflects phylogenetic conservatism in pest–host associations [49–51] and suggests that the addition of tree species to a community that are close relatives to a pest's preferred hosts may be more likely to amplify than dilute a pest. These dynamics, however, can be complicated by pests' lifecycle. Pests that rely on a particular sequence of hosts for successful infection (e.g., heteroecious rusts), or are not transmitted uniformly between hosts (for example, when insect vectors have feeding preferences and vary in their abilities to transmit pathogens [52]), may respond differently to diversity than those pests without strict host requirements. In these cases, pests response to diversity may be more impacted by host co-occurrence and, potentially, their interactions. Such a dynamic would complicate pest management as it implies that increasing diversity within forest stands could reduce the prevalence of some pests, while simultaneously increasing the prevalence of others. Pine-oak rust (*C. quercuum*) provides one such example as it is often managed by removing oaks around pine plantations in the southern US [53]. This native fungus, which alternates between pines and oaks, cannot complete its life cycle on pines alone, so removing oaks (and leaving only pines) can be an effective management strategy to reduce disease risk on economically important pines. This strategy, however, could increase the risk of pests that specialize on pine and reach their highest prevalence in pine monocultures.

## Implications for forest biodiversity

The influence of biodiversity on pest prevalence has important implications for understanding the ecological effects of shifting forest compositions and biodiversity. We demonstrate a generally strong dilution effect of forest biodiversity across a broad taxonomic spectrum of tree pests in tens of thousands of forest plots, and suggests that declining forest diversity could translate to higher pest levels, potentially increasing extinction risk of susceptible tree host species. Experimental manipulations further indicate that the magnitude of the dilution effect may be even greater following losses of biodiversity than estimates derived from space for time substitutions across other diversity gradients (e.g., latitudinal gradients) [54]. Pest pressure

could also feedback to impact forest diversity. For example, specialist pests in forest stands where tree species diversity is low and abundance is high, could act to suppress ecologically dominant species, and open the forest to other, non-host tree species. Over time, this process could increase tree diversity (c.f. Janzen–Connell hypothesis [55]), while simultaneously making the forest more resistant to pest invasion.

Critically, we show that pest responses to future biodiversity change will be contingent on the identity of the tree species lost and gained, whereby some pests could show dramatic increases in prevalence with reduced forest biodiversity and others remain largely unaffected by forest tree loss. While longer-term data would be needed to parse the influence of pests on biodiversity gains and losses, our results nonetheless indicate that low diversity forests, whether due to natural or human-caused processes, will on average likely experience greater and more frequent pest emergence events.

## Supporting information

**S1 Fig. Spatial autocorrelation of (a) modeled metric of pest prevalence, and (b) model residuals.** Each panel shows a random draw of 1,000 plots used to calculate spatial autocorrelation. Significance is determined by a two-sided permutation test, and indicates $p < 0.025$ or $p > 0.975$. Note autocorrelation is not significant for most distances in (b), indicating that the model is accounting for much of the spatial pattern. Data underlying this figure can be found at doi.org/10.6084/m9.figshare.21317979.
(TIFF)

**S2 Fig. Structural equation model used to predict pest prevalence.** The dataset used to fit this model was identical to the one used for the Bayesian model, but allowed for effects between diversity, temperature and precipitation, and diversity, evolutionary distance, and host abundance. Numbers show standardized effect sizes. Data underlying this figure can be found at doi.org/10.6084/m9.figshare.21317979.
(TIF)

**S3 Fig. Relationship between pest's median slope estimates with diversity and slopes with host abundances.** Pests are colored by general pest type; fungi include fungi-like organisms. See also Fig 2 and S4. Data underlying this figure can be found at doi.org/10.6084/m9.figshare.21317979.
(TIFF)

**S4 Fig. Slope estimates between host frequency and pest occurrences for all individual pests from our Bayesian regression model.** Negative values indicate a tendency towards amplification, and positive values indicate a tendency towards dilution. Points are colored by pest nativity. Points are medians, shaded segments are 80% credible intervals, and thinner lines are 95% intervals. See also Fig 3. Data underlying this figure can be found at doi.org/10.6084/m9.figshare.21317979.
(TIFF)

**S5 Fig. Conditional relationship between relative pest occurrence and the evolutionary distance between pest hosts and the tree communities where pests occur for (a) mean effect across all pests and (b) 4 individual pests.** In general, *S. clavigignenti-juglandacearum* and *M. disstria* were the pests most diluted by tree diversity, while *L. dispar* and *D. frontalis* were minimally affected by tree diversity (see Figs 3 and 4). Data underlying this figure can be found at doi.org/10.6084/m9.figshare.21317979.
(TIFF)

**S1 Table. Summary of Bayesian regression model of pest abundance.** Credible intervals for parameter estimates are given for 2.5 (2.5% CI) and 97.5 (97.5% CI) percentiles. Bulk effective sample size (ESS) indicates the efficiency of sampling in the bulk of the distribution around the mean, while tail effective sampling size indicates sampling in the tails of the distribution. All variables were scaled to mean of 0 and standard deviation of 1.0 before inclusion in the model.
(XLSX)

## Acknowledgments

The authors thank E. Wolkovich and A. Prasad for comments on an earlier draft of this manuscript.

## Author Contributions

**Conceptualization:** Andrew V. Gougherty, T. Jonathan Davies.

**Formal analysis:** Andrew V. Gougherty.

**Funding acquisition:** T. Jonathan Davies.

**Methodology:** Andrew V. Gougherty, T. Jonathan Davies.

**Writing – original draft:** Andrew V. Gougherty.

**Writing – review & editing:** Andrew V. Gougherty, T. Jonathan Davies.

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
