## [Editor Report · Decision Letter 0]

18 Apr 2023

Dear Dr Gougherty, 

Thank you for submitting your manuscript entitled "Host evolutionary history amplifies the dilution effect of biodiversity" for consideration as a Research Article by PLOS Biology.

Your manuscript has now been evaluated by the PLOS Biology editorial staff, as well as by an academic editor with relevant expertise, and I'm writing to let you know that we would like to send your submission out for external peer review.

Once your full submission is complete, your paper will undergo a series of checks in preparation for peer review. After your manuscript has passed the checks it will be sent out for review. To provide the metadata for your submission, please Login to Editorial Manager (https://www.editorialmanager.com/pbiology) within two working days, i.e. by Apr 20 2023 11:59PM.

Kind regards,

Roli Roberts

Roland Roberts, PhD

Senior Editor

PLOS Biology

rroberts@plos.org

---

## [Decision Letter · Decision Letter 1]

6 Jun 2023

Dear Dr Gougherty,

Thank you for your patience while your manuscript "Host evolutionary history amplifies the dilution effect of biodiversity" was peer-reviewed at PLOS Biology. It has now been evaluated by the PLOS Biology editors, an Academic Editor with relevant expertise, and by two independent reviewers. 

In light of the reviews, which you will find at the end of this email, we would like to invite you to revise the work to thoroughly address the reviewers' reports.

As you will see below, both reviewers are broadly supportive of your study, but make a number of suggestions to improve the manuscript, including, for example, additional analyses and experiments to increase the mechanistic insight.

Given the extent of revision needed, we cannot make a decision about publication until we have seen the revised manuscript and your response to the reviewers' comments. Your revised manuscript is likely to be sent for further evaluation by all or a subset of the reviewers.

**IMPORTANT - SUBMITTING YOUR REVISION**

*Re-submission Checklist*

*Published Peer Review*

*PLOS Data Policy*

*Blot and Gel Data Policy*

Sincerely,

Roli Roberts

Roland Roberts, PhD

Senior Editor

PLOS Biology

rroberts@plos.org

REVIEWERS' COMMENTS:

Reviewer #1:

General comments. This paper presents a nice system for evaluating relationships between disease and host communities. The finding that evolutionary relationships help predict the dilution effect is not surprising, but welcome. Indeed it is at odds with popular conclusions in mammal communities that tend to focus on competence rather than relatedness. However, the study has flaws in logic, design and interpretation similar to many dilution effect studies. Some of these can be avoided with improvements to the analysis and more careful interpretation.

Specific comments.

119. An odd feature to the data are that only three damaging agents per tree are reported…What would happen if there were 7 damaging agents? Which agents are included? How does the exclusion affect your results? Looking at your data it seems that >3 agents would be rare, but with the prevalence data and some assumptions about independent assortment, you could get close to assessing how commonly this was the case and then argue why it is not likely to be a problem. 

Line 123. It is not enough to cite other sources for the limitations of the data. The limitations relative to your study are critical to present to the readers of this paper.

135: If non-host damage is treated as error, how frequent were such errors? What do they say about the dataset in general?

140: how confident are you in the host range assessments from the literature? These are often highly subject to sampling error and bias. Why not use host range assessments from the data set instead? I would have more faith in those and they seem more relevant to the study. At the least, do it both ways.

Line 150. Incidence or prevalence. Incidence implies a per-time aspect of infection. I think you mean prevalence.

Line 150. I am all for Bayesian hierarchical models when applied to nested data (and Stan is the right choice), but Stan is hard for others to reproduce and you need to justify why you have used them. Also, why was an SEM approach not used? Finally, it is essential that others be able to reproduce your results. I see there is a promise to provide some of the data through Figshare and that other data are publicly available. Having writing a fair amount of R-Stan code, I know that links to data are not enough to reproduce the results. So please provide clear access to the data and annotated code used to produce the figures / key results. An R-Markdown file is an efficient way to do this. 

Line 153, 204 and 240. Although the dilution effect literature largely ignores it, it is critical to separate the effects of abundance from diversity. Any model trying to explain disease prevalence should include host density as the primary hypothesized driver. Residual effects of diversity may indicate a dilution effect. Furthermore, in an SEM, there may be indirect effects of density on diversity making the two difficult to separate. Line 204 implies you considered tree abundance as a driver, but it is critical to separate direct effects of abundance from diversity as well as the indirect effects of diversity on abundance. It is not entirely clear to me what was done. In my view, an SEM would be the right approach. To that end, to understand the diversity dilution effect, most models require assessing the assumption that communities are additive or substitutive. Dilution is more likely in substitutive communities. Given that your paper aims to explain variation in the dilution effect among sites, I would suggest that it is necessary to consider variation in additive vs substitutive effects among forest types. In short, please clarify the additive vs substitutive characteristics of the study system and carefully separate out the direct and indirect effects of density from "diversity".

Line 160. There are many diversity metrics. And they measure different things and often generate different results. Past dilution studies have cherry picked metrics that fit their expectations rather than those that fit predictions from theory. Justify your choices of metrics better. And, indicate to the reader how your results would have differed if you had used other common metrics (and why).

WRT metrics, you should also consider pathogen diversity as a function of forest diversity. The predictions there are pretty straightforward (host diversity begets parasite diversity). It would be easy for you to assess this association with your data at the plot level.

Line 162. BTW, this is a great way to express "diversity" (far better than species counts). You might want to highlight this distinction for the reader as well as the motivation for using it and how it applies (and does not apply to the dilution effect). In doing so, think carefully about evolutionary distance at different taxonomic groupings. E.g., ferns and trees are really different. And a forest with a lot of ferns would be more diverse. But you don't consider ferns because you focus only on tree health. This is just to say that your "diversity" is limited in scope. Explain why and how it might affect your results.

Line 169. I agree that is it good to control for climate, but latitude and altitude are also possible joint drivers of disease and forest diversity. These variables must be considered as drivers, ideally through an SEM framework.

Line 171. I would not consider -.61 to be a weak covariance. Moderate perhaps.

Line 176. If I understand correctly, you are considering the effect of tree diversity on the prevalence of a particular pest. Please also consider the effect of tree diversity on the combined prevalence of all pests. This is critically important if you want to discuss how diversity affects forest health (which is the effect of all pests combined, not the average effect of each pest individually).

Line 271. I think this grossly oversimplifies the Phytophtora story. It is an important system to discuss, but evidence for dilution is strongly sensitive to how one measures diversity. Please look into this a bit more.

Line 275. Although you might not have information about competence, you likely do have information on growth rates. Most dilution literature in the animal system speaks to the following: low diversity systems have weedy competent hosts, whereas diverse systems have more well-defended long lived hosts. I.e., in animal systems, the current views are more about correlations between diversity and competence rather than on diversity per se. I don't necessarily agree that this is a rule, but it could be measured in your system. Please explore as this explanation competes with your interpretation of evolutionary distance.

Line 285. Thanks for bringing up the pine-oak rust example. It is a nice example of how complex life cycle parasites might have different relationships to diversity. Yet your data do not consider this variable. Why not?

Line 299. This speculation is common in dilution effect studies. It is part of the negativity bias in conservation biology and is logically flawed. For instance, pathogen prevalence is not necessarily related to extinction risk. That greatly depends on the pathogen and host. And one could argue that a higher pathogen prevalence in low diversity stands is a mechanism that helps maintain forest diversity (Janzen-Connell hypothesis). For your concluding paragraph to be supported, you would have to show that rare species suffer higher impacts from pathogens in low diversity forests. I encourage you to investigate and report on this particular potential for feedbacks between pathogens and host diversity in your data. Otherwise, please avoid the negativity bias.

Reviewer #2:

The authors present the results of an analysis of the effects of a suite of variables on the occurrence of pests and pathogens of trees, using a database of tree health and composition available from the US Forest Service. They find strong evidence for dilution effects for many pests/pathogens, with Shannon diversity reducing prevalence, temperature increasing prevalence, and average phylogenetic distance of trees from the preferred host decreasing prevalence. 

This is a generally well-written manuscript that utilizes a comprehensive database of forest pests and pathogens to address an important concept. The magnitude of the data is excellent - dozens of pests and pathogens from thousands of forest locations. Generally, I suggest that the authors make the work more scholarly, particularly around their exploration of prior work on dilution effects. That said, my comments are addressable. This paper is phenomenological rather than mechanistic, and the authors could expand their exploration of potential underlying mechanisms.

One major area that could be improved during revision is on the response variable. I believe they are using the percent of trees infected, which would be prevalence, but they are not entirely clear on this point. E.g. in the abstract, they refer to "pest occurrences." In the Methods, e.g. line 150ish, they say that they modeled pest incidence, as the number of attacked hosts per pest within plots, standardized by total tree frequency. I'm not sure I could recreate this value from this information. How did they standardize total tree frequency? This is important because Rosenthal et al. and others have found a decline in percentage of infected hosts, while getting a different result if non-hosts are included. 

The authors intermingle results for pests (e.g. insects) and pathogens (e.g. viruses and fungi) here, without making distinction between them. Some prior work has been done looking at this distinction, I believe by Dave Civitello, and they should explore this here. 

Line 63 - "There is" could be rewritten to make a stronger start, e.g. "Better understanding the role of biodiversity in xxx is an urgent need."

Line 70 - This isn't accurate. All recent meta-analyses I can think of have supported the overall presence of dilution effects, whether for plants or animals. Authors of these meta-analyses often focus on their subtle differences, e.g. stronger for fungal pathogens than viral ones, or stronger for temperate than tropical systems. But there is, unequivocally, consensus that there are dilution effects and that they often occur. The three citations the authors indicate here are not formal meta-analyses, and at least one of them (Lafferty and Wood) is decidedly out of date in this fast-moving field. Their claims in this section are critically important to their paper. I encourage them to be more scholarly here, by citing the recent overview of the dilution effect by Ostfeld and Keesing for eample, and also by focusing on meta-analyses rather than narrative explorations of evidence. Here, they cite one paper about the diversity of managed forests from 2006. In this section, they should cite general (current) literature on dilution effects. If they then want to specialize on results from forests, they should do so more thoroughly. For example, there's a recent paper by Lisa Rosenthal that is relevant to their results, and they should explore the literature on forest pests/pathogens in real detail once they establish the general support for dilution. Also, generally, it's fine to say that dilution effects don't always occur, but not accurate to say we don't know when they will or won't - at this point there are lots of studies that show them stronger in some types of systems than others, etc. 

Line 76 - Start with something other than "It is likely…", e.g. "Tree pest responses are likely influenced…"

Line 195 - rephrase for clarity: "Infestation with a particular pest was reported on average (median) in 99 (±1140.56 SD) of the total 26,343 plots."

Line 236 - Here, some references supporting the statement would be useful. Also, the "but" here is strange because the Liu et al. paper strongly supports dilution effects across plants, based on their meta-analysis. I assume the "but" is because they find that dilution is stronger in some instances than others, but that's a subtle point here that either needs to be explored in more detail or omitted in favor of the overall finding of their study.

Line 238 - Here, cite references supporting the claim about host abundance. I assume they mean Rosenthal et al.?

Figure 3 - Here, "bolder lines" is confusing. Please edit to say shaded segments, or something like that. 

Figure S3 - In this figure, panel a has a non-zero y-axis, which dramatizes the effect, but which is also somewhat misleading. It should be re-drawn with zero, as in panel b.

---

## [Decision Letter · Decision Letter 2]

6 Dec 2023

Dear Dr Gougherty,

Thank you for your patience while we considered your revised manuscript "Host evolutionary history amplifies the dilution effect of biodiversity" for publication as a Research Article at PLOS Biology. This revised version of your manuscript has been evaluated by the PLOS Biology editors, the Academic Editor and the original reviewers.

Based on the reviews, we are likely to accept this manuscript for publication, provided you satisfactorily address the following data and other policy-related requests.

IMPORTANT - Please address the following:

a) Please could you change your Title to something more explicit for our broader readership? We suggest "The evolutionary history of host trees amplifies the dilution effect of biodiversity on forest pests," which clarifies the nature of the organisms studied.

b) Please provide a blurb, according to the instructions in the submission form.

c) Please address my Data Policy requests below; specifically, we need you to supply the numerical values underlying Figs 1, 2, 3, 4, S1AB, S3, S4, S5AB, either as a supplementary data file or as a permanent DOI’d deposition, perhaps as part of your existing Figshare deposition.

d) Please cite the location of the data clearly in all relevant main and supplementary Figure legends, e.g. “The data underlying this Figure can be found in S1 Data” or “The data underlying this Figure can be found in https://figshare.com/s/56925a31df28fce26562”

e) Please make any custom code available, either as a supplementary file or as part of your Figshare deposition (I see that the latter already contains R script for model fitting).

We expect to receive your revised manuscript within two weeks. 

*Published Peer Review History*

*Press*

Sincerely,

Roland

Roland Roberts, PhD

Senior Editor,

rroberts@plos.org,

PLOS Biology

DATA POLICY:

Regardless of the method selected, please ensure that you provide the individual numerical values that underlie the summary data displayed in the following figure panels as they are essential for readers to assess your analysis and to reproduce it: Figs 1, 2, 3, 4, S1AB, S3, S4, S5AB. NOTE: the numerical data provided should include all replicates AND the way in which the plotted mean and errors were derived (it should not present only the mean/average values).

CODE POLICY

Per journal policy, as the code that you have generated is important to support the conclusions of your manuscript, we require that you make it available without restrictions upon publication. Please ensure that the code is sufficiently well documented and reusable, and that your Data Statement in the Editorial Manager submission system accurately describes where your code can be found.

DATA NOT SHOWN?

REVIEWERS' COMMENTS:

Reviewer #1:

Thanks for your professional responses to my previous comments.

Reviewer #2:

This revised manuscript presents results of a survey of the effects of forest tree diversity on forest pest prevalence. The authors have revised the manuscript based on previous reviews. Here, they find an overall dilution effect, such that forest tree diversity reduces the prevalence of pests, but this effect is not present for all pests, nor is it equally strong when it does occur. They find that the effect is stronger when host trees in the forest are more distantly related to the preferred host species. 

This is a well-written manuscript, and the authors have done an exemplary job of addressing concerns from the previous version. They have also taken great care to address the concerns of the other reviewer. For example, they now include an SEM, and they have included host abundance. (As an aside, and to address the concerns of the other reviewer, changes in abundance of hosts as diversity changes are one of the possible mechanisms of dilution effects. They are not, as has been pointed out for two decades now, the most interesting mechanism, but that doesn't discount that they are a real mechanism linking changes in diversity to changes in pest prevalence.) 

An excellent manuscript with a large dataset and an important result that will likely be of broad interest.

---

## [Editor Report · Decision Letter 3]

14 Dec 2023

Dear Dr Gougherty,

Thank you for the submission of your revised Research Article "Evolutionary history of host trees amplifies the dilution effect of biodiversity on forest pests" for publication in PLOS Biology. On behalf of my colleagues and the Academic Editor, Anurag Agrawal, I'm pleased to say that we can in principle accept your manuscript for publication, provided you address any remaining formatting and reporting issues. These will be detailed in an email you should receive within 2-3 business days from our colleagues in the journal operations team; no action is required from you until then. Please note that we will not be able to formally accept your manuscript and schedule it for publication until you have completed any requested changes.

Sincerely, 

Roli Roberts

Senior Editor

PLOS Biology

rroberts@plos.org